# Health-related quality of life and associated factors among cancer patients in Ethiopia: Systematic review and meta-analysis

**Tadele Lankrew Ayalew** [1]*, **Belete Gelaw Wale**[1], **Kirubel Eshetu Haile**[1], **Bitew Tefera Zewudie**[2], **Mulualem Gete Feleke**[1]

**1** Department of Nursing, School of Nursing, College of Health Science and Medicine Wolaita Sodo University, Sodo, Ethiopia, **2** Department Nursing, College of Health Science and Medicine, Wolkite University, Wolkite, Ethiopia

* tadelelankrew@gmail.com

## Abstract

### Introduction

Cancer is the main cause of morbidity and mortality in every part of the world, regardless of human development. Cancer patients exhibit a wide range of signs and symptoms. Being diagnosed with cancer has a variety of consequences that can affect one's quality of life. The term "health-related quality of life" refers to a multidimensional concept that encompasses a person's whole health. The availability of data on the prevalence of poor quality of life among cancer patients in Ethiopia is critical in order to focus on early detection and enhance cancer treatment strategies. In Ethiopia, however, there is a scarcity of information. As a result, the aim of this study was to determine the pooled estimated prevalence of quality of life among cancer patients in Ethiopia.

### Materials and methods

This systematic review and meta-analysis were searched through MEDLINE, Pub Med, Cochrane Library, and Google Scholar by using different search terms on the prevalence of health-related quality of life of cancer patients and Ethiopia. Joanna Briggs Institute Meta-Analysis of Statistics Assessment and Review Instrument was used for critical appraisal of studies. The analysis was done using STATA 14 software. The Cochran Q test and $I^2$ test statistics were used to test the heterogeneity of studies. The funnel plot and Egger's test were used to show the publication bias. The pooled prevalence of health-related quality of life of cancer with a 95% confidence interval was presented using forest plots.

### Results

A total of 12 studies with 3, 479 participants were included in this review and the overall pooled estimates mean score of health-related quality of life among cancer patients in Ethiopia was 57.91(44.55, 71.27, $I^2$ = 98.8%, p≤0.001). Average monthly income (AOR:3.70;95%CI:1.31,6.10), Stage of cancer (AOR:4.92;95% CI:2.96,6.87), Physical functioning(AOR:4.11;95%CI:1.53,6.69), Social functioning(AOR:3.91;95% CI:1.68,6.14)

**Data Availability Statement:** All relevant data are within the paper.

**Funding:** The authors received no specific funding for this work.

**Competing interests:** The authors have declared that no competing interests exist.

**Abbreviations:** CDC, Centers for Disease Control and Prevention; HRQOL, health related quality of life; SNNPR, Southern Nation, Nationalities and Peoples of Region; WHO, World Health Organization.

were significantly associated with quality of life. Subgroup meta-analysis of health-related quality of life of cancer patients in Ethiopia done by region showed that a higher in Addis Ababa 83.64(78.69, 88.60), and lower in SNNP region16.22 (11.73, 20.71), and subgroup analysis done based on the type of cancer showed that higher prevalence of health-related quality of life among cancer patients was breast cancer 83.64(78.69, 88.60).

## Conclusion

This review showed that the overall health related quality of life was above an average. Furthermore, average monthly income, cancer stage, physical, and social functioning were all significant determinants in cancer patients' QOL.as a result, this review suggests that quality of life evaluation be incorporated into a patient's treatment routine, with a focus on linked components and domains, as it is a critical tool for avoiding and combating the effects of cancer and considerably improving overall health. In general, more research is needed to discover crucial determining elements utilizing more robust study designs.

## Introduction

Cancer is a leading cause of morbidity and mortality across the globe in every world region and irrespective of the level of human development [1]. Cancer is the second leading death worldwide behind cardiovascular disease [2]. Cancer is emerging as a formidable challenge in low-income countries that have limited logistics supply to protect the quality of life of citizens [3]. In developing countries, the burden of cancer overlaps with the magnitude of communicable diseases including HIV/AIDS, hepatitis virus, and human papillomavirus, which can contribute to the pathogenesis of cancer [4, 5]. Cancer is caused by both external factors (tobacco smoking, chemicals, radiation, and infectious organisms) and internal factors (inherited mutations, hormones, and immune conditions). These causal factors may act together or in sequence to initiate or promote carcinogenesis. The development of most cancers requires multiple steps that occur over many years. Certain types of cancer can be prevented by eliminating exposure to tobacco and other factors that initiate or accelerate this process. Other potential malignancies can be detected before cells become cancerous or at an early stage when the disease is most treatable [6, 7]. People living with cancer develop a variety of symptoms [8, 9]. Being diagnosed with cancer certainly has different sequelae which hamper their quality of life [4, 10, 11]. Health-related quality of life is a multidimensional concept concerning a person's general health conditions. It is a national representative tool for cancer survivors and examines lifestyle characteristics. It consists of domains related to social functioning, emotional, mental, and physical well being which are impaired in cancer patients. Following early screening and treatment of cancer, patients have certainly improved as a person's view of life, satisfaction, and pleasure with their life, cancer survivors still face many challenges, including long-term complications of treatment that hamper their health-related quality of life [4, 11, 12]. Studies showed that cancer career patients experience a poor health-related quality of life than the general population [13, 14]. Understanding the comprehensive prevalence health-related quality of life of cancer patients is vital to provide supplementary information for healthcare workers, improve approaches to care, modify therapies, and provide supportive care for the duration of the illness and enhance the quality of life of cancer patients.

Cancer is one of the diseases that has a global impact on patients' health-related quality of life. On the other hand, health-related quality of life is a technique used to assess the treatment outcome in cancer patients individually [1, 15]. Estimating the prevalence of a patient's health-related quality of life helps health care providers and policymakers assess the success of cancer management and intervention.

In impoverished nations, such as Ethiopia, cancer is one of the leading causes of morbidity and mortality. To our knowledge, this is the first meta-analysis of its sort in Ethiopia to assess the pooled prevalence of health-related quality of life among all types of cancer patients. As a result, the findings of this study will assist health care providers in maintaining cancer patients' health-related quality of life in addition to their pharmacological therapy. The findings of this review are useful in developing a strategy for focusing on the most important areas of cancer patients' health-related quality of life. After treating cancer patients, the healthcare workers measure work effectiveness by different methods. Collective information on the prevalence of health-related quality of life among cancer patients in Ethiopia is to be considered vital to focus on early diagnosis and improve the treatment method of cancer. Health-related quality of life is one of the tools used to measure self-perceived approaches to evaluate patients' views of their health status. It is assessed by a standard structured questionnaire called the quality of life questionnaire prepared by the European Organization for Research and Treatment of Cancer. However, there are scarce data in Ethiopia. Therefore, the present review aimed to assess cancer patients' health-related quality in Ethiopia.

## Materials and methods

### Search strategy and review process

This systematic review and meta-analysis was conducted from published researches on the prevalence of health-related quality of life among cancer patients in Ethiopia. The studies were retrieved through internet search from the databases of MEDLINE, PubMed, Cochrane Library, and Google Scholar. We used the 2020 Preferred Reporting Items for Systematic and Meta-analysis (PRISMA) protocol to estimate the prevalence of health-related quality of life among cancer patients in Ethiopia [16]. We checked the database (http://www.library.UCSF.edu) and the Cochrane library to ensure this had not been done before and to avoid duplication. We also checked whether there was any similar ongoing systematic review and meta-analysis in the PROSPERO database and there had been no previous similar studies undertaken in Ethiopia.

The search was done using the following search terms; prevalence, health-related quality of life among cancer patients, and Ethiopia. The reference lists of already identified studies were screened to retrieve articles. All published articles up to October 10, 2021, were included in this review.

### Eligibility criteria

**Inclusion criteria.**   Studies were included in this review of the study;

1. Participants: Included participants who are living in Ethiopia.

2. The design was cross-sectional, cohort, case-control, etc.

3. Was conducted on health-related quality of life among cancer patients in Ethiopia.

4. Was published in English.

5. Study design: All observational study designs (cross-sectional, case-control, and cohort) were included.

6. Setting: Studies were only conducted in Ethiopia.

7. Study: All studies (published and unpublished) that were published in the form of journal articles, master's thesis, and dissertations until the final date of data analysis were included.

8. Language: Only the English language was considered in this study.

**Exclusion criteria.**   We excluded articles that were not fully accessible after at least two email contact with the principal authors.

## Operational definition

**Health related quality of life.**   Is defined as an individual's satisfaction with their physical, psychological, social relationships, environment, and spiritual aspects of life, and it is one of the major factors in assessing the health and health-related well-being of cancer patients It usually used to measure in chronic conditions and frequently impaired to a great extent of the patients [17].

**Comorbid disease.**   A chronic disease with a confirmed diagnosis of a disease other than cancer disease [18].

**Alcohol intake.**   Individuals consume more than three units of alcohol per day [19].

## Quality assessment and data collection

Joanna Brings Institute Meta-Analysis of Statistics Assessment and Review Instrument (JBI--MAStARI) was used for critical appraisal of the study.

Joanna retrieved studies and was assessed for inclusion using their title and abstracts. Then a full review of articles for the quality of assessment was done before selecting for the final review. The details of studies that met the inclusion criteria were imported into the Joanna Briggs Institute's System for the Unified Management, Assessment and Review of Information (JBI SUMARI, The Joanna Briggs Institute, Adelaide, Australia) critical appraisal tools to evaluate the quality of all studies [20]. All authors independently assessed the article title and abstract for inclusion in the review based on established article selection criteria, appraising the quality of the studies by criteria adapted for reporting prevalence data and cross-sectional studies. Studies were considered low risk if a score of 7 and above on the quality assessment indicators (Table 1). Any discrepancy which arose between the reviewers were in the review process was solved through discussion with other reviewers.

## Data extraction

The data extraction was done using a tool developed by the 2014 Joanna Brings Institute Reviewers' Manual data extraction form by three authors (TL, BT, and MG) [20]. The data extraction tool includes information on the title, author, year of study, publication year, study design, sample size, study participants, study area, response rate, and the proportion of health-related quality of life among cancer patients in Ethiopia. Articles that fulfilled the predefined criteria were used as a source of data for the final analysis. The reviewers cross-checked it to ensure consistency. Any discrepancy was solved through discussion with other authors and the procedure was repeated to overcome the difference which resulted during extracting every single study.

**Table 1. Critical appraisal results of eligible studies in the systematic review and meta-analysis on the prevalence of health-related quality of life among cancer patients in Ethiopia, 2021.**

| S.no | Author | Q1 | Q2 | Q3 | Q4 | Q5 | Q6 | Q7 | Q8 | Q9 | Total |
|------|--------|----|----|----|----|----|----|----|----|----|-------|
| 1 | Hassen A., et al [11] | Y | Y | Y | N | Y | Y | Y | Y | Y | 8 |
| 2 | Ayana B., et al [22] | Y | N | Y | Y | U | Y | Y | Y | Y | 7 |
| 3 | Baraki A., et al [23] | Y | Y | Y | Y | N | Y | Y | Y | Y | 8 |
| 4 | Erku D., et al [24] | Y | Y | Y | Y | Y | Y | Y | Y | Y | 9 |
| 5 | Koboto D., et al [25] | Y | N | Y | Y | U | Y | Y | Y | Y | 7 |
| 6 | Ababa A., et al [26] | Y | Y | Y | Y | Y | Y | Y | Y | Y | 9 |
| 7 | Gebretekle G., et al [27] | Y | N | Y | Y | Y | Y | Y | Y | Y | 8 |
| 8 | Tadele N [28] | Y | Y | Y | Y | Y | Y | Y | Y | N | 8 |
| 9 | Aberaraw R., et al [29] | Y | Y | Y | Y | Y | Y | Y | Y | Y | 9 |
| 10 | Sibhat S., et al [30] | Y | Y | Y | Y | U | Y | Y | Y | Y | 8 |
| 11 | Abegaz T., et al [4] | Y | Y | Y | N | Y | Y | U | Y | Y | 7 |
| 12 | Zeleke N., et al [31] | Y | Y | Y | Y | Y | Y | Y | Y | Y | 9 |

Y = Yes, N = No, U = Unclear; JBI Critical Appraisal Checklist for Studies Reporting

## Methodological quality assessment of studies

The methodological quality of included studies was appraised using a modified and predefined checklist to assess the methodological quality aspect quality of life among cancer patients reported.

A score of yes was given for an item if meeting the methodological criteria. A score of no was given for an item is not meeting the methodological criteria, and if an item neither met the criteria nor described the related parameter sufficiently was give unclear. Here, we use the above terms to screen the eligible articles for systematic review and meta-analysis. The point is that both "No and neither yes or no" articles illegible for this systematic review and meta-analysis. Here the JBI-MAStARI requires for the use as a methodological tool, specifically for assessing risk of bias with 9 items modified quality of life assessment checklist. According to Newcastle-Ottawa quality assessment Scale (NOS) score 7 or more for cross- sectional studies was accepted. Based this a score of 7 or more out of 9 acceptable for this review [21].

List of questions to assess the methodological quality of studies on QOL of cancer patients

Q1 = was the sample frame appropriate to address the target population?

Q2. Were study participants sampled appropriately?

Q3. Was the sample size adequate?

Q4. Were the study subjects and the setting described in detail?

Q5. Was the data analysis conducted with sufficient coverage of the identified sample?

Q6. Were the valid methods used for the identification of the condition?

Q7. Was the condition measured in a standard, reliable way for all participants?

Q8. Was there appropriate statistical analysis?

Q9. Was the response rate adequate, and if not, was the low response rate managed appropriately?

### Outcome measurement

Quality of life among cancer patients is the primary outcome of this review and meta-analysis. Mean is the summary measure of this outcome.

### Publication bias and heterogeneity

The existence of heterogeneity was assessed by using the funnel plot test, $I^2$, and its corresponding p-value. A value of 25%, 50%, and 75% was used to declare the heterogeneity test as low, medium, and high heterogeneity. For results with statistically significant heterogeneity, a random effect model of analysis was used. Egger regression asymmetry test was used to assess the statistical significance of publication bias [25].

### Data analysis

The data were entered using Microsoft Excel. The Meta-analysis was conducted using Stata 14 software. Forest plots were used to present the combined estimate with the 95% confidence interval (CI). The estimated pooled prevalence of health-related quality of life among cancer patients in Ethiopia was computed with 95% CI. Subgroup analysis was done by region, year of the study period, and study participants. Additionally, a univariate meta-regression model was applied by taking the sample size, publication year, and quality scores of each primary study to investigate the sources of heterogeneity. Finally, a forest plot figure was used to present the point proportions with their 95% CI of the primary studies. The heterogeneity of included studies was evaluated with $I^2$ statistics. Based on $I^2$ statistics, a value less than 25% were considered low heterogeneity, between 50 and 75% medium heterogeneity and greater than 75% were considered as high heterogeneity [32].

## Results

### Studies identified

A total of 3583 articles were retrieved through internet searching. One hundred six were identified through other sources. A total of 3,666 articles were retrieved. Out of these, 201 duplicate records were removed from the review. Of the total articles, 3,125 were due to inaccuracy title and 253 articles were due to absence similarity of abstracts were excluded from the review. After a full review of the articles, 74 were excluded by eligibility criteria. Finally, twelve studies were included in this meta-analysis (Fig 1).

### Characteristics of included studies

This systematic review and meta-analysis included 12 articles with 3,479 study participants. All studies employed a cross-sectional study design. Most of the regions in Ethiopia were represented in this review. Seven studies were from Addis Ababa city, four studies from the Amhara region, and two from SNNRP, region respectively. Studies were conducted from 2015 to 2020. The sample size of studies ranged from 140 [22] to 403 in a study conducted in Addis Ababa [11], and the response rate ranged from 93% to 100%. Overall, this review included a total of 3,483 health-related quality of life among cancer patients in Ethiopia (Table 2).

### Health related quality of life measurements

A wide range of health related quality of life evaluation measurements were utilized in these 12 revised articles. The findings of the health related quality of life measures were employed in each reviewed articles (Table 3).

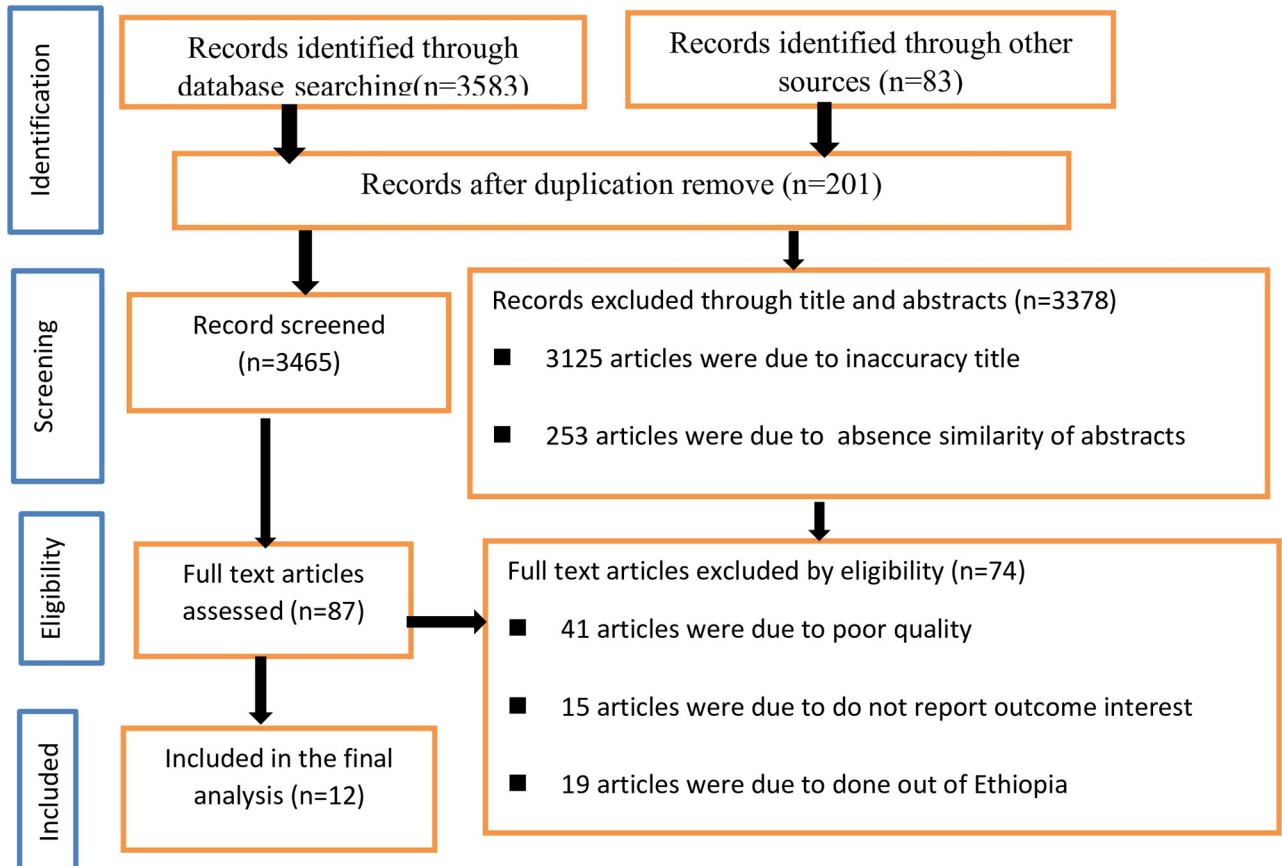

**Fig 1. PRISMA diagram of selecting and including studies for a systematic review and meta-analysis for the prevalence of health-related quality of life of cancer patients in Ethiopia, 2021.**

## The prevalence of health-related quality of life among cancer patients in Ethiopia

In Ethiopia, the prevalence of health-related quality of life among cancer patients in this review was high. A prevalence of health-related quality of life among cancer in SNNRP region 16.22% [25] in Addis Ababa 83.64% [29] were observed. The $I^2$ test result showed high heterogeneity ($I^2$ = 98.8%, p-value≤0.001) which is indicative to use a random-effects model of analysis. Therefore, using the random effect analysis, the overall pooled prevalence of health-related quality of life among cancer patients in Ethiopia was 57.91(44.55, 71.27, $I^2$ = 98.8%, p≤0.001) (Fig 2).

The funnel plot is symmetrical, the observation of the pooled prevalence of health-related quality of life among cancer patients was not affected by publication bias. The egger regression asymmetry test also demonstrated that no statistically significant publication bias may occur in this review (Egger's test, b = 0. 026, p ≤ 0.938) (Fig 3).

## Subgroup analysis

Subgroup meta-analysis of this review was done by using region and publication year showed that a higher pooled health-related quality of life among cancer patients was present when a

**Table 2. Characteristics of studies included in a systematic review on the prevalence of health-related quality of life among cancer patients in Ethiopia, 2021.**

| Author | Year | Region | Study area | SD | Study period | Measurement tool | Type of ca | Sample size | Cases | Prevalence HRQOL |
|---|---|---|---|---|---|---|---|---|---|---|
| Hassen A., et al [11] | 2019 | AA | TASH | CS | Feb-Apr 2018 | EORTC QLQ-C30 | breast cancer | 403 | 214 | 52.98 |
| Ayana B., et al [22] | 2018 | AA | TASH | CS | Jan-June 2014 | EORTC QLQ-C30 | Cervical cancer | 140 | 57 | 40.95 |
| Baraki A., et al [23] | 2020 | Amhara | FHRH | CS | Apr-May 2019 | PHQ-9 | All ca | 302 | 214 | 70.86 |
| Erku D., et al [24] | 2016 | Amhara | UOGTRH | CS | Oct 2015 -Febr 2016 | EORTC QLQ-C30 | All ca | 154 | 122 | 79 |
| Koboto D., et al [25] | 2020 | SNNPR | HU | CS | Apr- June 2019 | WHOQOL-BREF | Breast Cancer | 259 | 42 | 16.1 |
| Ababa A., et al [26] | 2020 | AA | TASH | CS | Mar-May 2017 | CQOLC | All ca | 291 | 239 | 82.23 |
| Gebretekle G., et al [27] | 2016 | Amhara | UOGTRH | CS | Jan-June 2014 | EORTC QLQ-C30 | All ca | 395 | 312 | 78.9 |
| Tadele N [28] | 2020 | AA | TASH | CS | Jan-June 2018 | EORTC QLX245D | cervical cancer | 379 | 183 | 48.3 |
| Aberaraw R., et al [29] | 2015 | AA | TASH | CS | Mar- May 2013 | EORTC QLQC30 | Breast cancer | 388 | 114 | 29.4 |
| Sibhat S., et al [30] | 2020 | AA | TASH | CS | Mar-Apr 2019 | EORTC-C30 | Breast cancer | 214 | 179 | 83.61 |
| Abegaz T., et al [4] | 2019 | AA | TASH | CS | Dec 2017-Feb 2018 | EORTC LBR235D | Breast cancer | 404 | 240 | 59.32 |
| Zeleke N., et al [31] | 2018 | Amhara | UOGTRH | CS | Jan-Aug 2017 | EORTC QLQ-C30 | All ca | 150 | 79 | 52.7 |

AA = Addis Ababa, CS = Cross-sectional, UOGTRH = University of Gondar teaching referral hospital, TASH = Tinkur Ambesa Specialized hospital, FHRH = Felege-Hiwot referral Hospital, PHQ-9 = patient health questionnaire-9, CQOLC = Caregiver Quality of Life Index-Cancer, EORTC QLQ-C30 and QLQBR23 = European organization for research and treatment of cancer core 30 and quality of life questionnaire specific to breast EORTC QLQ-CX24 and EQ-5D = European Organization for Research and Treatment of Cancer module (EORTC QLQ-C30), cervical cancer module (EORTC QLQ-CX24), and Euro Quality of Life Group's 5-Domain Questionnaires 5-Levels (EQ-5D) questionnaires

study done in Addis Ababa 83.64(78.69, 88.60). The lower prevalence health-related quality of life among cancer patients was observed in the SNNP region16.22 (11.73, 20.71) (Fig 4).

The subgroup analysis was done by using a type of cancer and the authors showed that the higher prevalence of health-related quality of life among cancer patients was breast cancer 83.64(78.69, 88.60) (Fig 5).

## Sensitivity analysis

To identify a single study influence on the overall meta-analysis, sensitivity analysis was performed using a random-effects model, and the results showed that there was no strong evidence for the effect of a single study on the overall meta-analysis result. The table shows that the estimate from a single study was closer to the combined estimate, which implies the absence of a single study effect on an overall study.

## Factors associated with health related quality of life among cancer patients in Ethiopia

Association between average monthly income and health-related quality of life. Five papers were included in the meta-analysis to determine the relationship between health-related quality of life and average monthly income [11, 22, 24, 28, 30]. This meta-analysis found that having an average monthly income of more than 4,000 ETB was significantly associated with health-related quality of life among cancer patients. Patients with an average monthly income of 4,000 ETB were 3.10 times more likely to acquire a good quality of life than those with a lower monthly income (OR = 3.10, 95% CI: 1.31–6.10). Because a fixed effect model was adopted, the included studies showed no heterogeneity ($I^2$ = 0.001%, p = 0.383) (Fig 6).

Association between physical functioning and health-related quality of life. The meta-analysis includes five studies that showed between physical functioning and health-related quality of life [4, 11, 22, 27, 28]. Accordingly, cancer patients with good physical functioning were 4.11

**Table 3. WHOQOL related finding of quality of life among cancer patients in Ethiopia.**

| Author | QOL measurement tool | Reported Domain | WHOQOL Related Findings |
|---|---|---|---|
| Hassen A., et al [11] | Selected items of EORTC QLQ-C30 and EORTC QLQ-BR23. | Overall QOL | Mean global health status (QOL) = 52.98 (SD = 25.61). |
| | | Psychological functioning | emotional functioning ((± SD) 47.61 ± 25.83) |
| | | | cognitive functioning ((± SD) 80.06 ± 22.89) |
| | | Social relations functioning | patients scored worse in sexual functioning (85.8%) |
| | | | only 16.6% participants had poor body image |
| Ayana B., et al [22] | EORTC QLQ-C30 | Overall QOL | Global Health Status score was 40.95(SD ± 24.35) |
| | | Physical | had higher score in fatigue (65.24, SD±22.59) |
| | | Role function | Active coping and religious coping were positively correlated with emotional well being (50.12, SD± 35.11) |
| | | Emotional function | Active coping and acceptance were positively correlated with functional well-being (55.48, SD±30.32) |
| | | Cognitive | had poor cognitive functioning (88.21, SD = 18.49) |
| | | Social function | had higher score in fatigue, pain, dyspepsia, financial difficulties and constipation (42.26, SD = 32.08) |
| Baraki A., et al [23] | PHQ-9 | Overall QOL | Patients with cervical cancer reported better QOL than patients with breast cancer. |
| | | Physical | Sleeping difficulties (44.3%). |
| | | Psychologica | Depression (27.8%) and loss of confidence (3.5%). |
| | | Independence | Inability to work (30%). |
| Erku D., et al [24] | Structured questionnaire and EORTC QLQ-C30 | Overall QOL | Excellent overall QOL (14%). |
| | | Physical | Pain (72%), lack of energy (78%), sleeping difficulties (63%). |
| | | Psychological | Loss of confidence (30.9%), difficulty in concentration (46%), |
| | | Independence | Difficulties in daily activities (75%). |
| | | Social | Social support (56.6–67.1%), sexual functioning (11.2%). |
| Koboto D., et al [25] | WHOQOL | Overall QOL | overall global health scale was 75.3 (SD±17.1) |
| | BREF | Environmental | 93.31 (SD±19.76) |
| | | Physical health | 88.26 (SD±21.61) |
| | | Psychological | 68.2 (SD±19.07) |
| | | Social related | 36.69 (SD±7.62) |
| Ababa A., et al [26] | CQOLC | Overall QOL | overall mean score of the QOL was 82.23 (±16.21). |
| | | burden | burden was 24.49 (±7.83), |
| | | disruptiveness | for disruptiveness was 16.63 (±5.69), |
| | | adaptation | for positive adaptation was 18.58 (±3.42), |
| | | financial | for financial concern was 9.29 (±3.27) |
| | | other | for other subscale scores was 12.94 (±4.18) |
| Gebretekle G., et al [27] | EORTCQLQ-C30 | Overall QOL | overall mean score of the QOL was (54.86±4.67) |
| Tadele N [28] | EORTC QLQ-CX24 | Global health status/ QoL | Mean global health statu scale scores were 48.3 ± 23.77, 0.77 |
| Aberaraw R., et al [29] | EORTC QLQC30 | GQOL | mean of global health status/QoL was 57.28 (SD = 25.28). |
| | | Physical | physical functioning had a mean of 62.71 (SD = 34.86). |
| | | Role | Role functioning had a mean of 43.36(SD = 43.32), |
| | | Emotional | emotional functioning had a mean of 45.88 (SD = 42.28) |
| | | Social | social functioning had a mean of 39.69(SD = 39.69) |
| Sibhat S., et al [30] | Selected items of EORTC QLQ-C30 and EORTC QLQ-BR23. | Overall QOL | Low overall QOL (mean: 48.25). |
| | | Physical Capacity | High level of postoperative breast symptoms (mean: 19.1) and arm symptoms (mean:24.5). |
| | | Psychological | High level of body image (mean: 69.3); Low level of perspective toward the future (mean: 40.3) |
| | | Social Relations | Low level of sexual functioning (mean: 85.3). |

*(Continued)*

**Table 3.** (Continued)

| Author | QOL measurement tool | Reported Domain | WHOQOL Related Findings |
|---|---|---|---|
| Abegaz T., et al [4] | EORTC QLQ-C30 | Overall QOL | Global health status 52.7 (20.1) |
| Zeleke N., et al [31] | | Physical | Physical functioning 53.27 (22.9) |
| | | Role | Role functioning 43.32 (26.7) |
| | | Social | Social functioning 46.31 (25.5) |
| | | Emotional | emotional functioning 61 (25.5) |
| | | Cognitive | Cognitive functioning 59.31 (43.6) |

QOL = Quality of life; FACT-G: Functional Assessment of Cancer Therapy-General; FACT-B: Functional Assessment of Cancer Therapy-Breast; EORTC QLQ-C30: European Organization for Research and Treatment of Cancer Quality of Life Questionnaire; EORTC QLQ-BR23: European Organization for Research and Treatment of Cancer Quality of Life Questionnaire-Breast cancer;

PHQ-9 = patient health questionnaire-9, CQOLC = Caregiver Quality of Life Index-Cancer

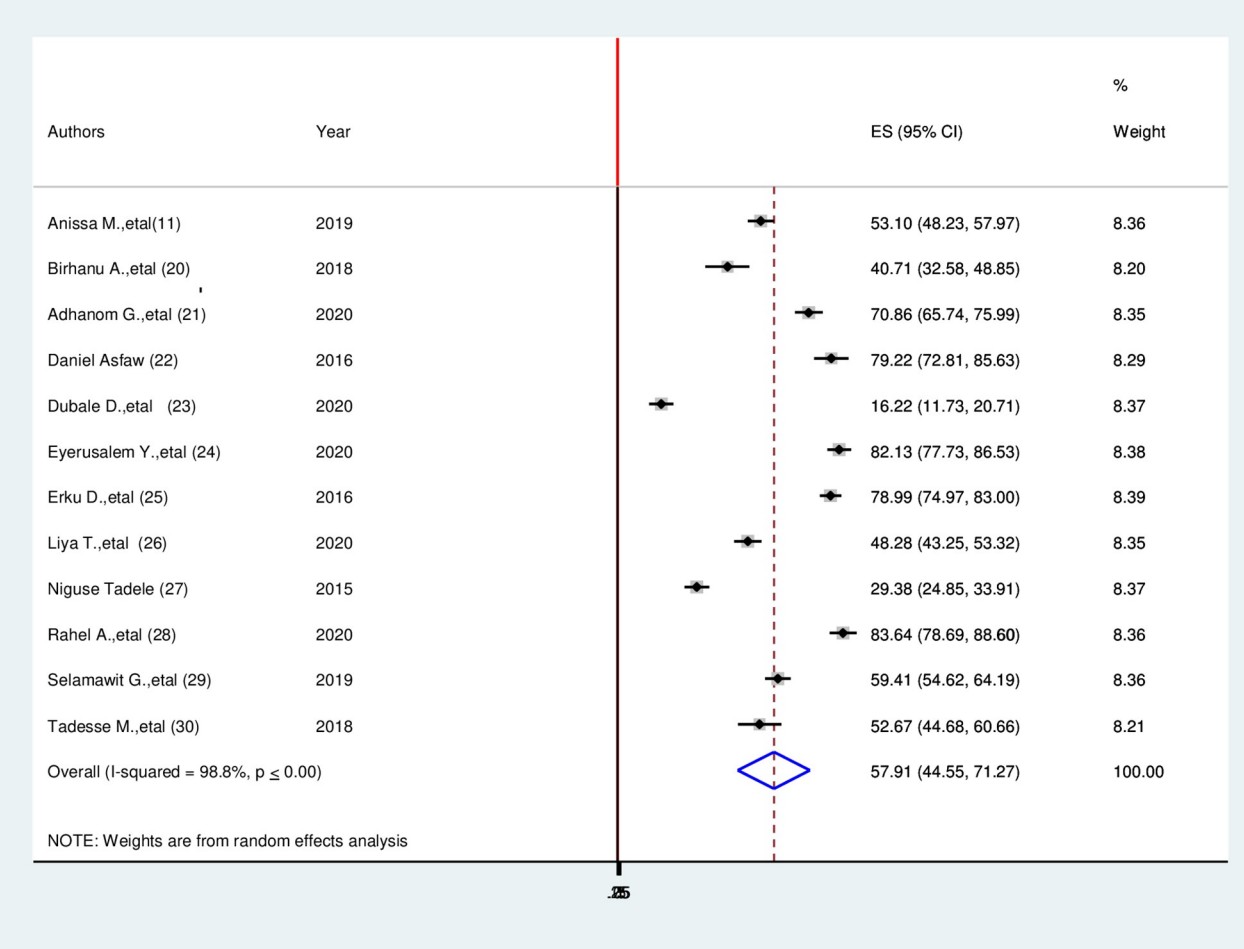

**Fig 2. Forest plot for the prevalence of health-related quality of life among cancer patients in Ethiopia, 2021.**

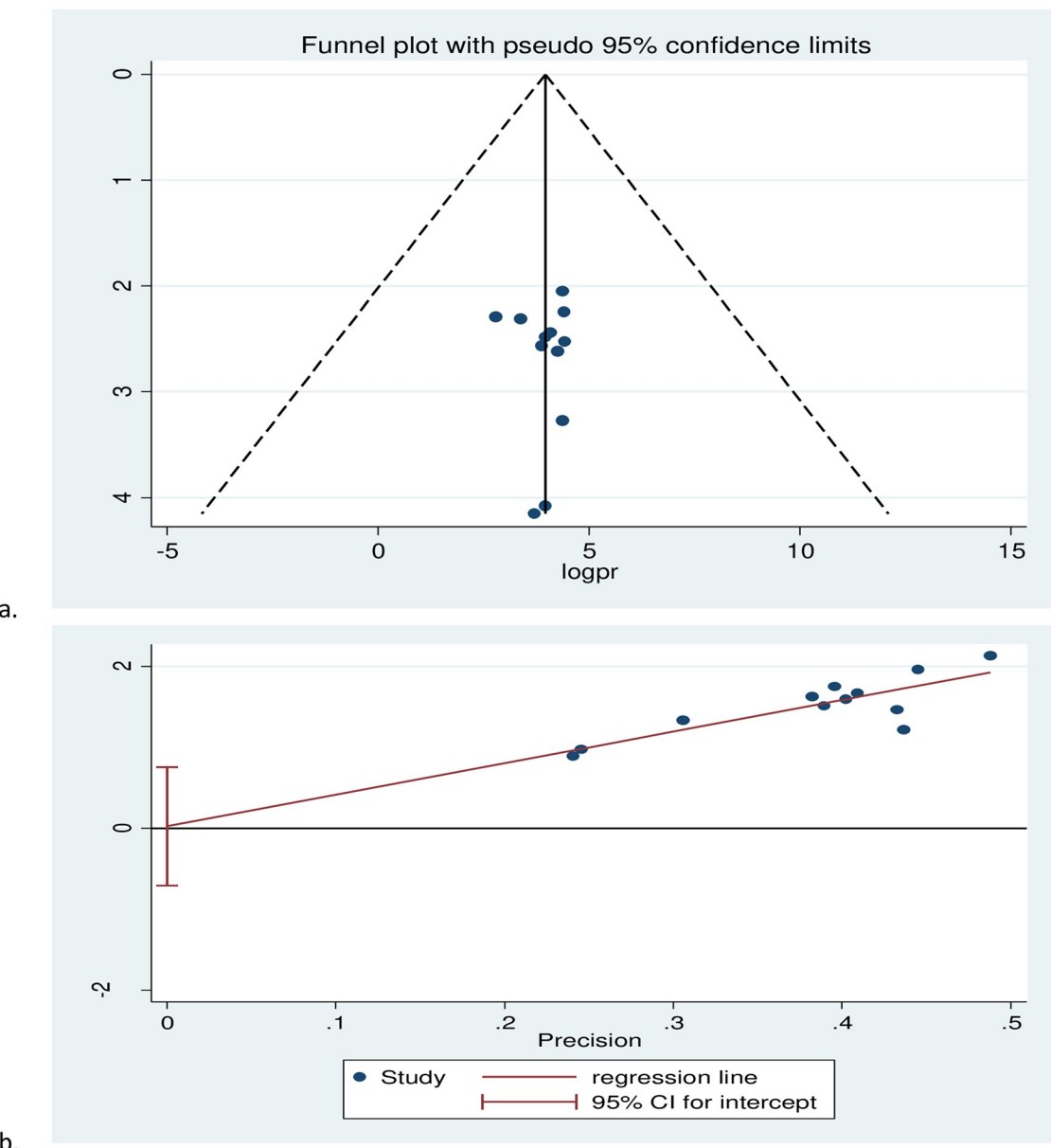

**Fig 3. Funnel plot (a) and Egger's test (b) of health related quality of life of cancer patients in Ethiopia, 2021.**

times more likely to develop good quality of life than cancer patients with low physical function (OR = 4.11, 95% CI: 1.53–6.69) (Fig 7).

Association between social functioning and health-related quality of life. To demonstrate the association between quality of life and social functioning, six studies were chosen [4, 11, 22, 27–29]. Cancer patients with social functioning were 3.91 times more likely than cancer patients with poor social functioning to develop good quality of life (OR = 3.91, 95% CI: 1.88–6.14) (Fig 8).

Association between stage of cancer and health related quality of life. This meta-analysis included seven research to show the association between quality of life and cancer stage among cancer patients [4, 24, 27–30, 33], and two of the included studies found statistical

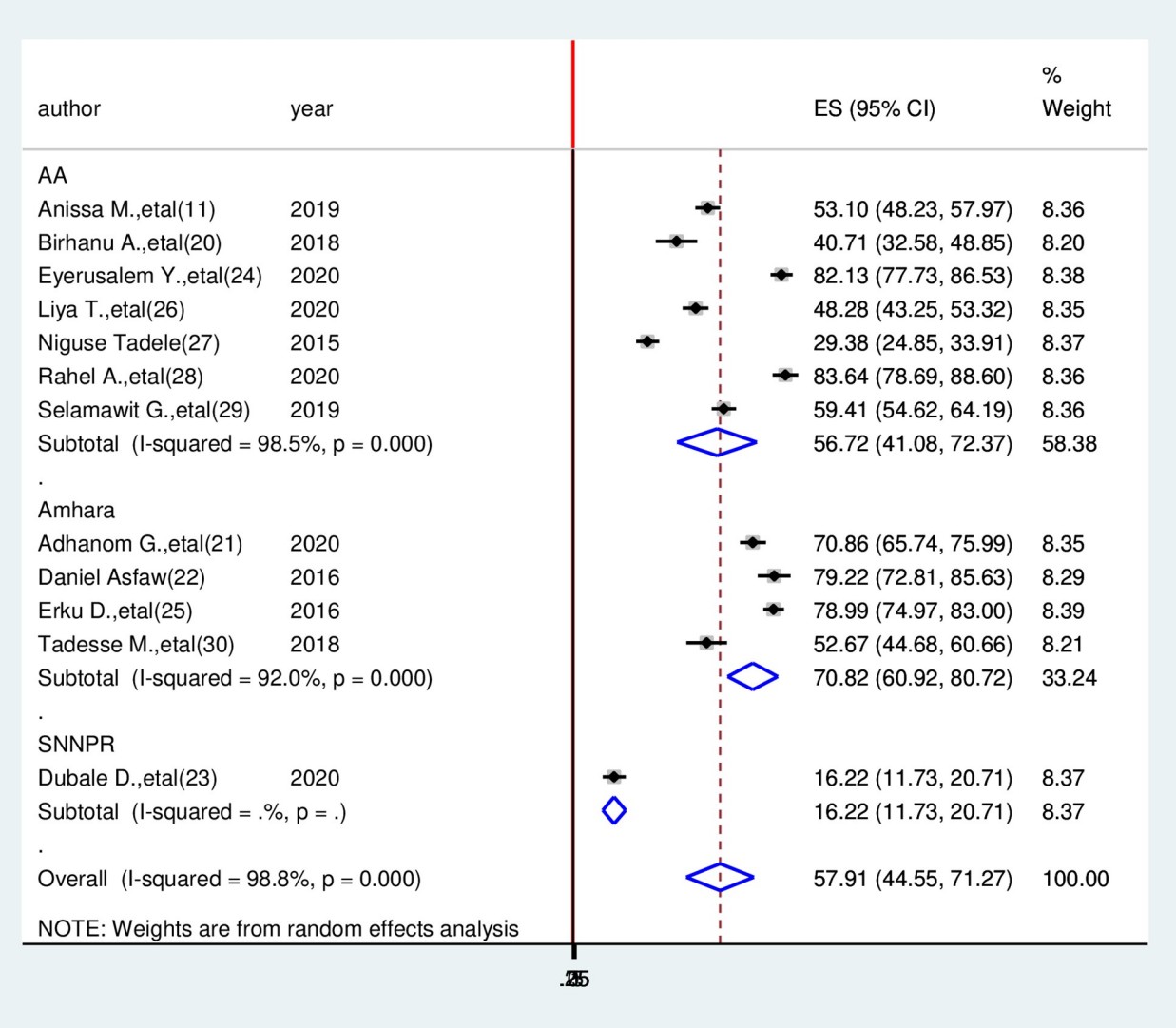

**Fig 4. The subgroup analysis done by region for the prevalence of health related quality of life among cancer patients, 2021.**

significance between quality of life and disease stage [29, 30]. This finding indicated that patients with stage four cancer had a 4.92 times higher chance of developing poor quality of life than those with others (OR = 4.92, 95% CI: 2.96–6.87). As a result of using a fixed effect model, the included studies did not show heterogeneity ($I^2$ = 0.00%, p = 0.483) (Fig 9).

## Discussion

The goal of this study was to determine the countrywide prevalence of cancer patients' health-related quality of life. In this study, the overall pooled prevalence of cancer patients' health-related quality of life was 57.91 (95% CI: 44.55, 71.27, $I^2$ = 98.8%, p≤ 0.001). The results of this meta-analysis are lower than those of Norwegian systematic reviews and meta-analyses [25, 31]. The possible reason could be attributable to differences in geographic location, medical service quality, and socioeconomic position.

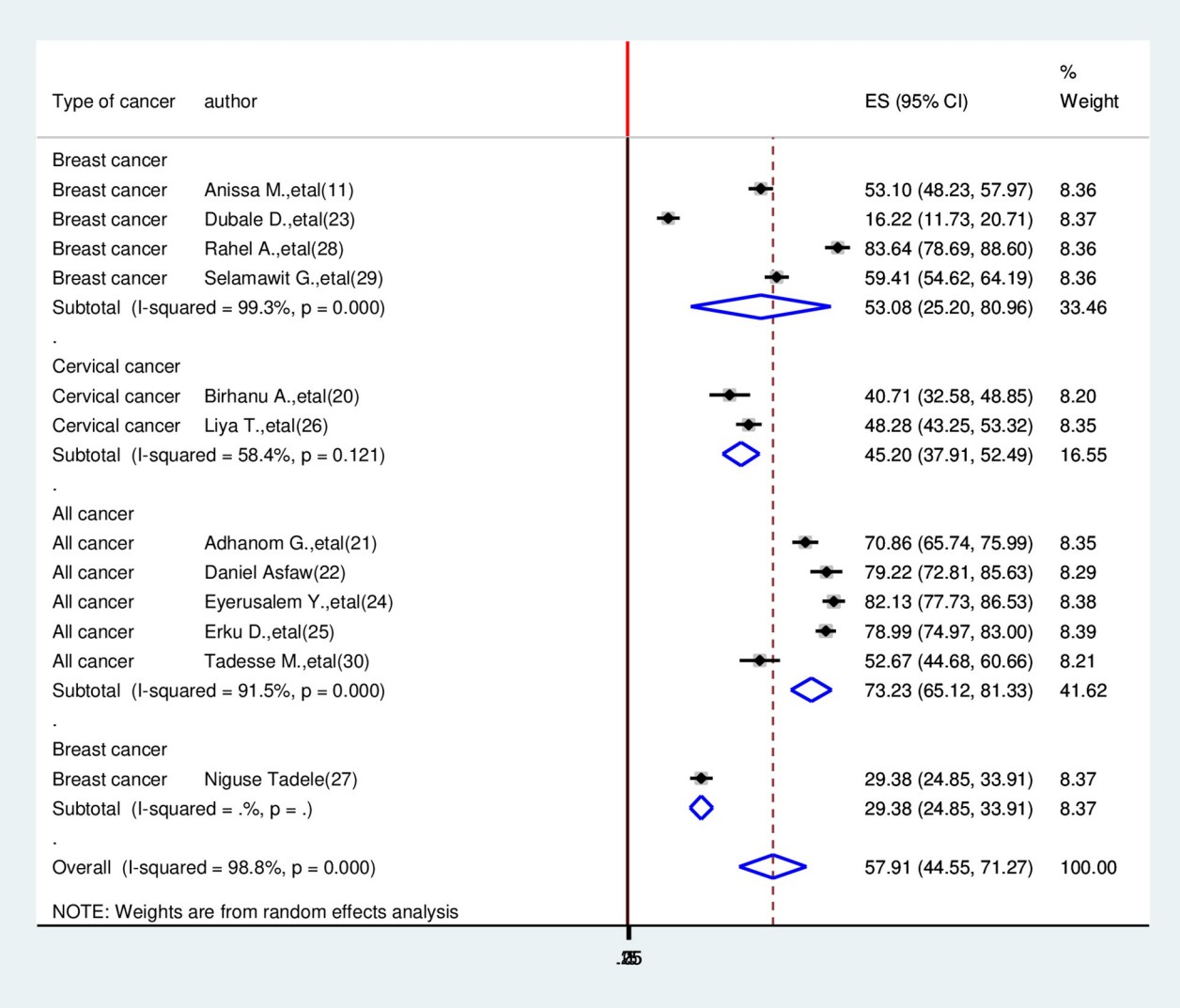

**Fig 5. The subgroup analysis done by type of cancer with the prevalence of health related quality of life among cancer patients, 2021.**

However, this result is significantly greater than that of research conducted in Bangladesh (28.67%), Uganda and Mozambique and the United States [32]. The possible explanation for the observed variations might be due to differences in methodology and sample size used to assess the prevalence of health-related quality of cancer patients by individual studies conducted in each country. Moreover, the difference could be due to the difference in a geographical area, quality of medical service, and socio-economic status which have an unlimited effect to assess the prevalence of health-related quality of life of cancer patients.

In the current systematic review and meta-analysis, several factors were associated with health related quality of life among cancer patients. Regarding to socio-demographic characterstics, high average monthly income was significantly associated with good health-related quality of life. This review is supported by a study done in Nigeria, Pakistan, and Ethiopia [11, 22, 24, 28, 30]. The possible reason may be adequate access of information

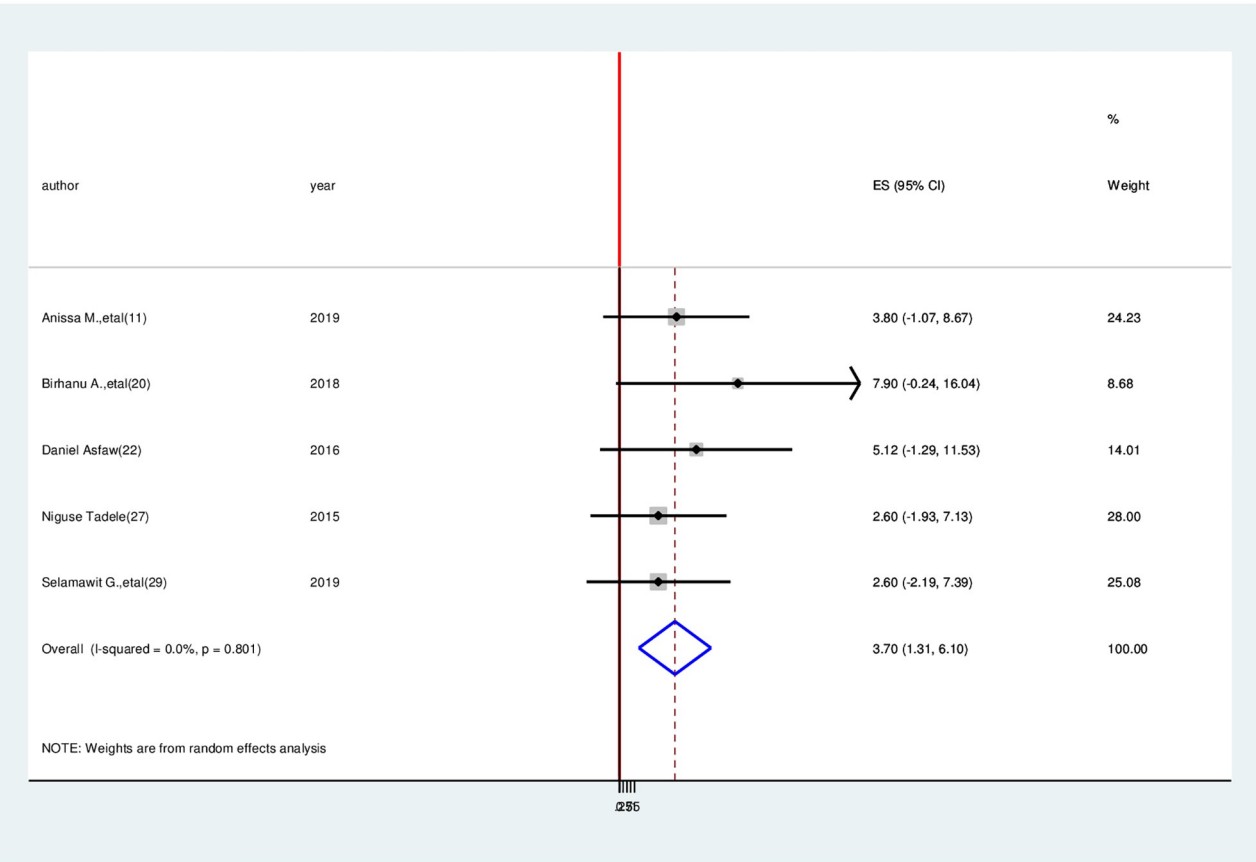

**Fig 6. Association between average monthly income and health-related quality of life among cancer patients in Ethiopia, 2021.**

about cancer patients that can help them to give self-adjustment for the patient and they are high likely to communicate with healthcare provider. Because High average monthly income has been associated to a variety of features of improved patient care, including being less concerned about financial troubles and missing work [17]. Similarly patients facing difficulties with monthly income are at risk for experiencing upset of their quality of life [33]. Furthermore, in countries such as Ethiopia, this is made worse by the fact that there are only a limited number of facilities available for chemotherapy treatment, requiring all patients to travel significant distances to receive treatment, adding to the patients' already high financial burden.

This comprehensive review and meta-analysis found that having good physical functioning helps cancer patients acquire a positive health-related quality of life. This finding is consistent with a prior study conducted in Ethiopia in various circumstances [34]. Poor physical functioning may increase the likelihood of poor quality of life due to a lack of daily activities and a lack of early care of any anomaly that may emerge in the daily foundation activities.

Social functioning was strongly associated with health related quality of life among cancer patients in Ethiopia. This review is supported with a prior studies [31, 35, 36]. The possible justification for this reason may be in Ethiopian societies, families, friends, relatives, and neighbors can all provide valuable social support. Furthermore, patients hold a strong religious belief and believe that if they receive good treatment, they will be cured.

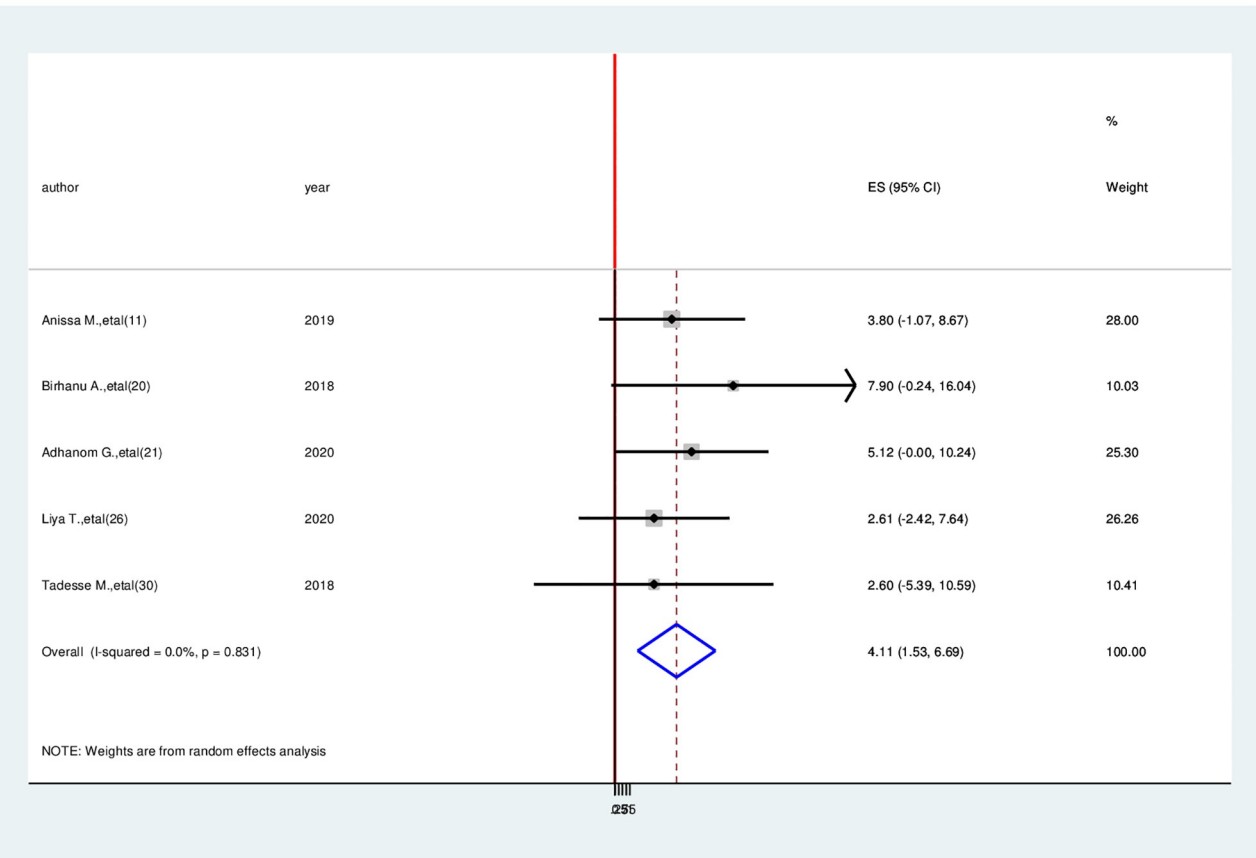

**Fig 7. Association between having physical functioning and health-related quality of life among cancer patients in Ethiopia.**

The findings of this review also revealed that the advanced stage of cancer disease and length of time a patient has been diagnosed with cancer is one of the risk factors for low quality of life in cancer patients. This review and meta-analysis was supported by studies done previously [31, 37]. As the stage of the disease increases, the likelihood of poor quality of life increases. This is owing to the fact that if the disease is not appropriately treated and controlled, it will worsen over time.

## Strengths of this review

The strength of this study includes the use of multiple databases to search articles (both manual and electronic search) and the uniform abstraction of material in a predetermined manner by two separate reviewers helped to minimize error. This meta-analysis also included studies from different parts of the country that comprise both urban and rural populations.

## Limitations

The number of studies included in this systematic review and meta-analysis was limited by the requirement that primary studies be written in English. This review was hampered by the fact

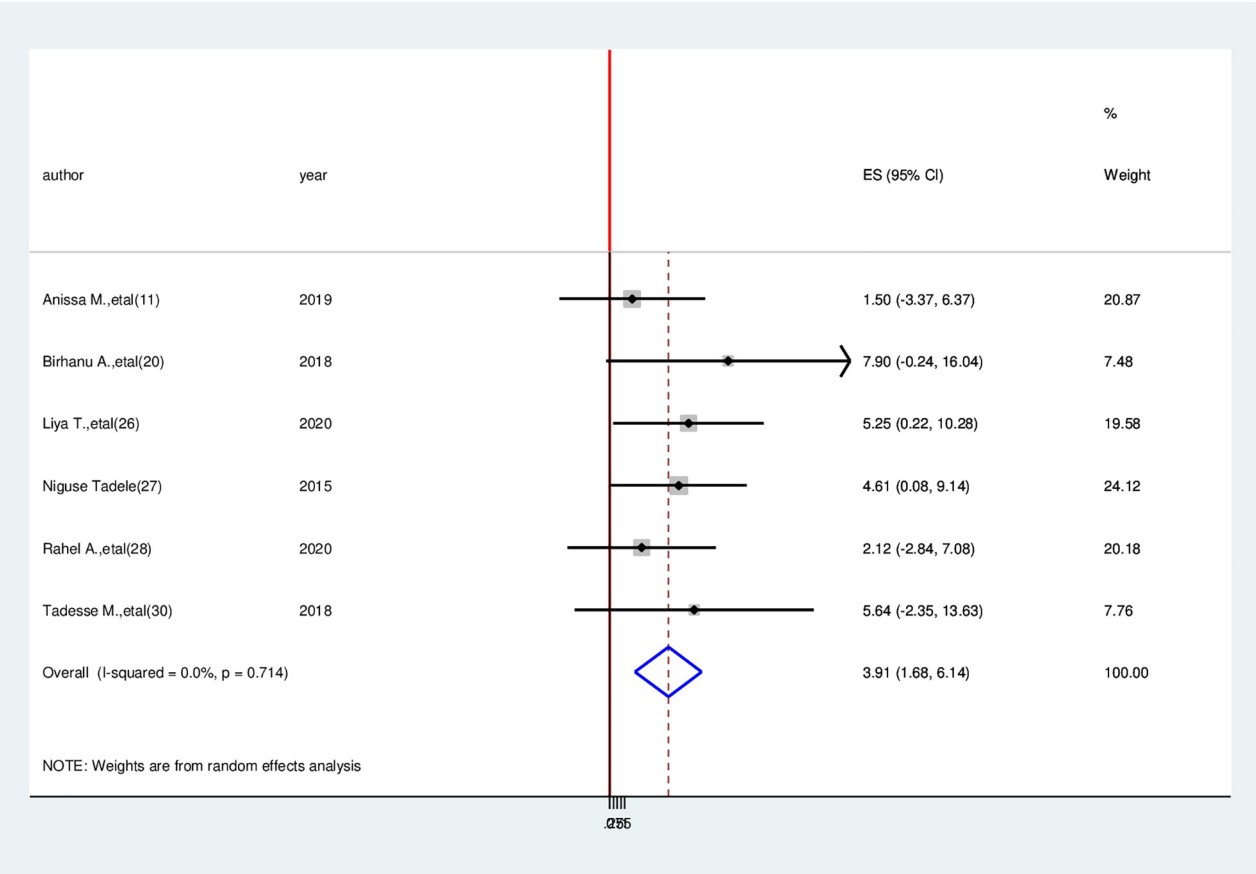

**Fig 8. Association between having social functioning and health-related quality of life among cancer patients in Ethiopia.**

that no primary articles were conducted that focused on cancer patients' age category, gender based comparison, or socioeconomic status. Another drawback of this review was the small sample size due to few primary articles were included in meta-analysis. Furthermore, all of the studies included in this review used a cross-sectional study design, which means that the outcome variable could be influenced by other confounding variables, lowering the study's power and making it more difficult to draw causal conclusions between associated factors and cancer patient quality of life.

## Conclusion

This review showed that the overall health related quality of life was above an average. Furthermore, average monthly income, cancer stage, physical, and social functioning were all significant determinants in cancer patients' QOL.as a result, this review suggests that quality of life evaluation be incorporated into a patient's treatment routine, with a focus on linked components and domains, as it is a critical tool for avoiding and combating the effects of cancer and considerably improving overall health. In general, more research is needed to discover crucial determining elements utilizing more robust study designs.

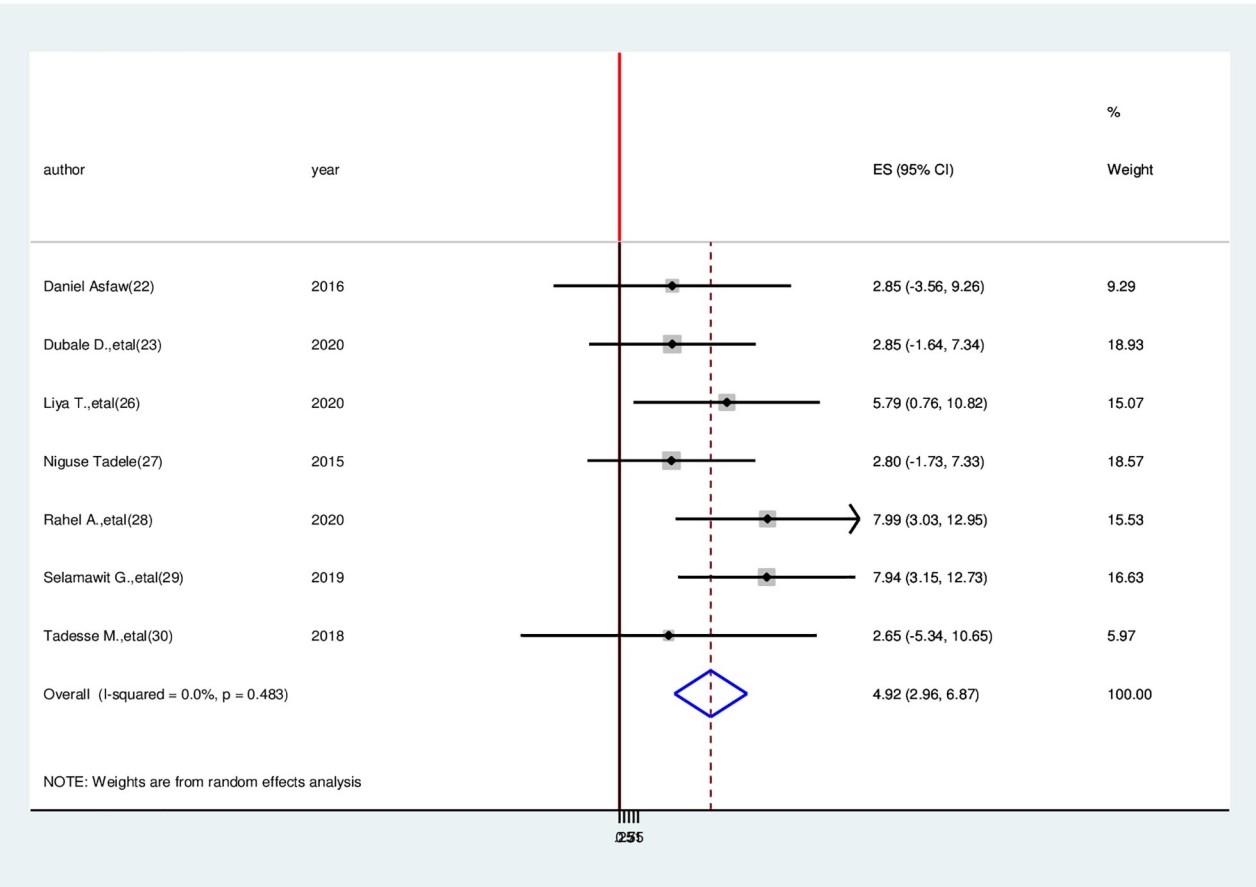

**Fig 9. Association between stage of cancer and health-related quality of life among cancer patients in Ethiopia.**

## Supporting information

**S1 File.**
(DOCX)

**S1 Checklist.**
(DOCX)

## Acknowledgments

We would like to acknowledge the authors of the studies included in this review.

## Author Contributions

**Conceptualization:** Tadele Lankrew Ayalew, Belete Gelaw Wale, Kirubel Eshetu Haile.

**Data curation:** Tadele Lankrew Ayalew, Belete Gelaw Wale.

**Formal analysis:** Tadele Lankrew Ayalew, Belete Gelaw Wale, Kirubel Eshetu Haile, Bitew Tefera Zewudie.

**Funding acquisition:** Tadele Lankrew Ayalew, Belete Gelaw Wale, Kirubel Eshetu Haile, Bitew Tefera Zewudie.

**Investigation:** Tadele Lankrew Ayalew, Belete Gelaw Wale, Bitew Tefera Zewudie.

**Methodology:** Tadele Lankrew Ayalew, Bitew Tefera Zewudie.

**Project administration:** Tadele Lankrew Ayalew.

**Resources:** Tadele Lankrew Ayalew.

**Software:** Tadele Lankrew Ayalew, Mulualem Gete Feleke.

**Supervision:** Tadele Lankrew Ayalew, Mulualem Gete Feleke.

**Validation:** Tadele Lankrew Ayalew, Mulualem Gete Feleke.

**Visualization:** Tadele Lankrew Ayalew, Mulualem Gete Feleke.

**Writing – original draft:** Tadele Lankrew Ayalew, Mulualem Gete Feleke.

**Writing – review & editing:** Tadele Lankrew Ayalew.

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
