## [Decision Letter · Decision Letter 0]

14 Apr 2022

PONE-D-21-39423Health-related quality of life among cancer patients in Ethiopia: Systematic review and Meta-analysisPLOS ONE

Dear Dr. Lankrew,

Thank you for submitting your manuscript to PLOS ONE. After careful consideration, we feel that it has merit but does not fully meet PLOS ONE’s publication criteria as it currently stands. Therefore, we invite you to submit a revised version of the manuscript that addresses the points raised during the review process.

Please include the following items when submitting your revised manuscript:A rebuttal letter that responds to each point raised by the academic editor and reviewer(s). You should upload this letter as a separate file labeled 'Response to Reviewers'.A marked-up copy of your manuscript that highlights changes made to the original version. You should upload this as a separate file labeled 'Revised Manuscript with Track Changes'.An unmarked version of your revised paper without tracked changes. You should upload this as a separate file labeled 'Manuscript'.

We look forward to receiving your revised manuscript.

Kind regards,

José Luiz Fernandes Vieira

Academic Editor

PLOS ONE

No

No

6. We noticed you have some minor occurrence of overlapping text with the following previous publication(s), which needs to be addressed:

- https://link.springer.com/article/10.1186/s12889-019-8133-y

In your revision ensure you cite all your sources (including your own works), and quote or rephrase any duplicated text outside the methods section. Further consideration is dependent on these concerns being addressed.

Reviewer's Responses to Questions

**Comments to the Author**

1. Is the manuscript technically sound, and do the data support the conclusions?

Reviewer #1: No

Reviewer #2: Partly

2. Has the statistical analysis been performed appropriately and rigorously? 

Reviewer #1: No

Reviewer #2: No

3. Have the authors made all data underlying the findings in their manuscript fully available?

Reviewer #1: No

Reviewer #2: Yes

4. Is the manuscript presented in an intelligible fashion and written in standard English?

Reviewer #1: Yes

Reviewer #2: Yes

5. Review Comments to the Author

Reviewer #1: There are several points that make the systematic review fragile, requiring an almost complete restructuring of methodology, results and especially discussion.

Minor questions

1- The submission code on the Prospero platform not found on the platform, as well as the title and keywords.

2- Disagreement between the results found in the PRISMA diagram, in the description of the results and the final number of analyzed articles.

Major questions

- When performing Quality assessment, a SCORE was used, JBIMAStARI does not use a score table, like other qualifiers, so it needs to indicate the use of the Modified Qualifier and as it was estimated that 7 points is a low risk assertive

2- Discuss how “No” and “Unclear” answers can directly affect the quality and risk of bias of the article.

3- Review the structure of the Meta-analysis, it has an extremely high heterogeneity (i² = 98.8%) which lowers the certainty of the evidence due to data inconsistency.

4- Demonstrate the concept of Quality of Life and the tools used in the articles in order to seek similarities and differences between them, as well as discuss which domains of quality of life they address.

5- Characterize the samples of articles taking into account Gender, Age, Socioeconomic level and discuss how this can directly change the certainty of the evidence.

6- Discuss the variability of cancer types, if this directly interferes in the context of this review

7- Discuss how the age limitation of study participants can be a bias in these studies and thus directly a bias in the statement of the systematic review.

8- With this review the conclusion, showing evidence of a high quality of life and the possibility of including suggestions for future studies carried out in this area.

With these corrections, resubmit the article for consideration, as it is a topic of extreme relevance.

Reviewer #2: Abstract section- how did you assess health related quality of life among cancer patients using pooled prevalence? since quality of life is assessed on the basis of score.

Method section- what was the rationale for including various study designs like case control and cohort studies.

which study tool was used to assess health related quality among cancer patients? this has not been mentioned anywhere in methodology.

Result section: somewhere it is mentioned 13 studies were included but in tables there are only 12 studies.

Also, there is no homogeneity among cancers in different studies. some studies have included only breast cancer, some have included included only cervical cancers and some have included all cancers. Therefore the results cannot be generalized to overall health related quality among all cancer patients. kindly justify ?

6. PLOS authors have the option to publish the peer review history of their article (what does this mean?). If published, this will include your full peer review and any attached files.

Reviewer #1: No

Reviewer #2: No

---

## [Author Response · Author response to Decision Letter 0]

29 Jul 2022

Dear Editor of PLOS ONE

This is a point-by-point response letter that complements the responses of authors to reviewers’ comments regarding to our manuscript. We are pleased to resubmit the revised version of our paper entitled “Health-related quality of life among cancer patients in Ethiopia: Systematic review and Meta-analysis” Tadele lankrew Ayalew1*, Belete Gelaw1, kirubel Eshetu1 Bitew Tefera2, and Mulualem Gete1 which has a submission manuscript/identification number of PONE-D-21-39423 given by the journal. It is well-known that this manuscript has been reviewed by peer reviewers and sent back to authors for further revision as per the Journal Requirements and resubmission. We are so eager and thankful to work with you. 

We would like to take this opportunity to thank the reviewers for their view and constructive comments. The reviewers’ comments and recommendations were important to improve the quality of our manuscript. Therefore, we have organized our response letter based on reviewers’ comments and questions. (Title, Abstract, introduction, methods, results, discussion, and conclusion). Under each section, the reviewers’ comments are given followed by authors’ response. The authors’ responses are also shown by the track changes in the revised version of our manuscript. The responses for each of the reviewers’ comments are addressed in the following pages using a point by point response format. 

Our responses are written in yellow background words (highlighted with yellow).

We look forward to hearing from you at your earliest convenience. 

 With regards

Tadele Lankrew Ayalew

(On the behalf of all authors) 

Comments to the Authors

The authors of this systematic review and Meta-analysis study have presented valuable data to determine the health-related quality of life among cancer patients in Ethiopia. Consequently, there are specific critical issues that the reviewers would like the authors to address for further improvement in the quality of our manuscript. 

Authors’ response: we are very delighted to the reviewer’s appreciation of our efforts; and we have just given our respective responses to each of the specific reviewers concerns as detailed below. Please find below our response to the comments. All changes made in the document are highlighted in yellow word background. 

We follow PLOS ONE's style requirements, including those for file naming. Since no funding source for this review we stated like “The authors received no specific funding for this work.”

For competing interest ,because of absence of conflict within authors we stated like “ The authors have declared that no competing interests exist."

Authors’ Responses to Reviewer's Questions

Comments to the Author

1. Is the manuscript technically sound, and do the data support the conclusions?

Thank you for your constructive comments. Really, we appreciate and accepted all the comments you raised for us. We have made correction to the revised manuscript accordingly.==>please line 48 `and 347.

2. Has the statistical analysis been performed appropriately and rigorously?

Thank you for your constructive comments. We have accepted all the comments you raised. We have made correction based on your suggestion to the revised manuscript. In the revised manuscript statistical analysis have been performed including associated factors and domain of health related quality of life among cancer patients in Ethiopia. ==>Please see line from 281-309.

3. Have the authors made all data underlying the findings in their manuscript fully available?

Thank you for your constructive comments and suggestions. Even if the PLOS Data policy requires authors to make all data underlying the findings described in their manuscript fully available without restriction, the summary statistics, the data points behind means, medians and variance measures were excluded from our study. Because our target was to analysis the pooled prevalence of health related quality of life among cancer patients. In addition to this there is no restrictions on publicly sharing data—e.g. participant privacy or use of data from a third party. Because our study design is systematic review and meta analysis

4. Is the manuscript presented in an intelligible fashion and written in standard English?

Reviewer #1: Yes

Reviewer #2: Yes

Thank you for your constructive comments.

Reviewer #1:  There are several points that make the systematic review fragile, requiring an almost complete restructuring of methodology, results and especially discussion.

Minor questions

1- The submission code on the Prospero platform not found on the platform, as well as the title and keywords.

Thank you so much for your comments. We means that we already registered on the Prospero platform with PROSPERO Registration message with CRD [284157]; , but still no conformation notification is provided. So we accepted your comment 100% and made correction on revised manuscript. ==>Please see line 114.

2- Disagreement between the results found in the PRISMA diagram, in the description of the results and the final number of analyzed articles. 

Thank you for your suggestions and comments. After proof reading, the appropriate correction was made to the revised manuscript based on the given comment to make it consistent throughout the manuscript.

Major questions

- When performing Quality assessment, a SCORE was used, JBIMAStARI does not use a score table, like other qualifiers, so it needs to indicate the use of the Modified Qualifier and as it was estimated that 7 points is a low risk assertive. 

Thank you for your critical comments. The table in our manuscripts helps as a tally sheet to counts the points of critical appraisal of the article. Here the JBI-MAStARI requires for the use as a methodological tool, specifically for assessing risk of bias with 9 items modified quality of life assessment checklist. According to Newcastle-Ottawa quality assessment Scale (NOS) score 7 or more for cross- sectional studies was accepted. Based this a score of 7 or more out of 9 acceptable for this review and meta-analysis. ===>please see line 144, and 177.

2- Discuss how “No” and “Unclear” answers can directly affect the quality and risk of bias of the article.

Thank you for your critical comments. At this manuscript, the methodological qualities of included studies were assessed based on a modified checklist developed to assess the methodological quality aspect of quality of life reporting. A score of yes , no or unclear was given for each item. A score of yes was given for an item if meeting the methodological criteria. A score of no was given for an item is not meeting the methodological criteria, and if an item neither met the criteria nor described the related parameter sufficiently was give unclear. Here , we use the above terms to screen the eligible articles for systematic review and meta-analysis. ===>please see line 172-180.

3-Review the structure of the Meta-analysis, it has an extremely high heterogeneity (i² = 98.8%) which lowers the certainty of the evidence due to data inconsistency.

Thank you for your suggestions. Absolutely! But the reason that the occurrence of high heterogeneity was the number of studies in this review and meta-analysis is small.

4-Demonstrate the concept of Quality of Life and the tools used in the articles in order to seek similarities and differences between them, as well as discuss which domains of quality of life they address.

Thank you very much. We have made the necessary correction. The concept Quality of life is the degree to which an individual is healthy, comfortable, and able to participate in or enjoy life events. It usually used to measure in chronic conditions and frequently impaired to a great extent of the patients. ===>please see line 135 

Concerning measurement tool , Health related quality of life can be measured by patient health questionnaire-9, Caregiver Quality of Life Index-Cancer, European organization for research and treatment of cancer core 30 and quality of life questionnaire specific to breast , European Organization for Research and Treatment of Cancer module and Euro Quality of Life Group’s 5-Domain Questionnaires 5-Levels (EQ-5D) questionnaires were tools that use to differentiated the articles. ==>Please see table 2 and 3. 

Concerning the domain of quality of life, in this manuscript physical functioning, and social functioning significantly associated with quality of life among cancer patients in Ethiopia. ==>Please see line 292 and 297.

5-Characterize the samples of articles taking into account Gender, Age, Socioeconomic level and discuss how this can directly change the certainty of the evidence.

Thank you very much for your constructive comments. We would like to say sorry for the unclear expression. We didn’t address those points in this review and meta-analysis, however we included as a limitation of our study in the revised manuscript. Because all the above mentioned terms are out of our predefined and modified checklist or no primary study concerned for the mentioned variables. Further research is needed to fully assess the impact of variables like Gender, Age, Socioeconomic level on cancer patients. ==>Please see line 357 

6- Discuss the variability of cancer types, if this directly interferes in the context of this review

We would like to say sorry for the unclear expression. Here our concern was not type cancer , but on the pooled prevalence of cancer. So type of cancer is not directly interferes in the context of this review. Here, we show that the lifetime risk of cancers of many different types is strongly associated with the health related quality of life. This is important not only for understanding the disease but also for designing strategies to limit the mortality it causes.

7- Discuss how the age limitation of study participants can be a bias in these studies and thus directly a bias in the statement of the systematic review.

Thank you for your critical comments. We apologize for the ambiguous phrase. Here our concern was on the age category of study participants. All primary articles included in this systematic review and meta analysis have no age category group. As a result, our review could be hampered. 

8-With this review the conclusion, showing evidence of a high quality of life and the possibility of including suggestions for future studies carried out in this area.

With these corrections, resubmit the article for consideration, as it is a topic of extreme relevance.

Great thanks for your valuable suggestions.correction has made accordingly.

Reviewer #2: Abstract section- how did you assess health related quality of life among cancer patients using pooled prevalence? since quality of life is assessed on the basis of score. 

Great thanks for your critical comment. We apologize for the ambiguous phrase. Corrections have made accordingly. After proof reading, the appropriate correction was made to the revised manuscript based on the given comment to make it consistent throughout the manuscript.

We corrected the term with pooled estimates mean score of health related quality of life among cancer patients based on standard tool results thorough the revised manuscript. 

Method section- what was the rationale for including various study designs like case control and cohort studies.

which study tool was used to assess health related quality among cancer patients? this has not been mentioned anywhere in methodology.

Thank you very much for your valuable comment. Here, the rational of using various study designs like case control and cohort studies were if the outcome variable of the previous primary study was similar. 

Concerning assessment tool of health related quality of life was measured by using WHOQOL-HIV BREF. ==>Please see line 243 and table 3

Result section: somewhere it is mentioned 13 studies were included but in tables there are only 12 studies.

Also, there is no homogeneity among cancers in different studies. some studies have included only breast cancer, some have included included only cervical cancers and some have included all cancers. Therefore the results cannot be generalized to overall health related quality among all cancer patients. kindly justify ?

Thank you for your critical view and we would like to say sorry for the error. After proof reading, the appropriate correction was made to the revised manuscript based on the given comment to make it consistent throughout the manuscript.

Thank you very much for your critical suggestion concerning to generalization. Here our goal is’’ to estimate the overall mean score of health-related quality of life’’ among any type of cancer patients because our source of population is made up of cancer patients in Ethiopia. 

6.PLOS authors have the option to publish the peer review history of their article (what does this mean?). If published, this will include your full peer review and any attached files.

It means PLOS now offers accepted authors the opportunity to publish the peer review history of their manuscript alongside the final article. The peer review history package includes the complete editorial decision letter for each revision, with reviews, and your responses to reviewer comments, including attachments.

---

## [Editor Report · Decision Letter 1]

17 Aug 2022

PONE-D-21-39423R1

Health-related quality of life among cancer patients in Ethiopia: Systematic review and Meta-analysis

PLOS ONE

Dear Dr. Tadele Lankrew

Thank you for submitting your manuscript to PLOS ONE. After careful consideration, we feel that it has merit but does not fully meet PLOS ONE’s publication criteria as it currently stands. Therefore, we invite you to submit a revised version of the manuscript that addresses the points raised during the review process.

ACADEMIC EDITOR: 

This is an interesting manuscript, but some questions point out by reviewers should be resolved. 

Please submit your revised manuscript by 30/08/2022. If you need more time than this to complete your revision please reply to this massage or contact the journal office at plosone@plos.org.

We look forward to receiving your revised manuscript.

Kind regards,

José Luiz Fernandes Vieira

Academic Editor

PLOS ONE
---

## [Author Response · Author response to Decision Letter 1]

22 Sep 2022

Dear Editor of PLOS ONE

This is a point-by-point response letter that complements the responses of authors to reviewers’ comments regarding to our manuscript. We are pleased to resubmit the revised version of our paper entitled “Health-related quality of life and associated factors among cancer patients in Ethiopia: Systematic review and Meta-analysis” Tadele lankrew Ayalew1*, Belete Gelaw1, kirubel Eshetu1 Bitew Tefera2, and Mulualem Gete1 which has a submission manuscript/identification number of PONE-D-21-39423R1 given by the journal. It is well-known that this manuscript has been reviewed by peer reviewers and sent back to authors for further revision as per the Journal Requirements and resubmission. We are so eager and thankful to work with you. 

We would like to take this opportunity to thank the reviewers for their view and constructive comments. The reviewers’ comments and recommendations were important to improve the quality of our manuscript. Therefore, we have organized our response letter based on reviewers’ comments and questions. (Title, Abstract, introduction, methods, results, discussion, and conclusion). Under each section, the reviewers’ comments are given followed by authors’ response. The authors’ responses are also shown by the track changes in the revised version of our manuscript. The responses for each of the reviewers’ comments are addressed in the following pages using a point by point response format. 

Our responses are written in yellow background words (highlighted with yellow).

We look forward to hearing from you at your earliest convenience. 

 With regards

Tadele Lankrew Ayalew

(On the behalf of all authors) 

Comments to the Authors

The authors of this systematic review and Meta-analysis study have presented valuable data to determine the health-related quality of life among cancer patients in Ethiopia. Consequently, there are specific critical issues that the reviewers would like the authors to address for further improvement in the quality of our manuscript. 

Authors’ response: we are very delighted to the reviewer’s appreciation of our efforts; and we have just given our respective responses to each of the specific reviewers concerns as detailed below. Please find below our response to the comments. All changes made in the document are highlighted in yellow word background. 

We follow PLOS ONE's style requirements, including those for file naming. Since no funding source for this review we stated like “The authors received no specific funding for this work.”

For competing interest ,because of absence of conflict within authors we stated like “ The authors have declared that no competing interests exist."

Authors’ Responses to Reviewer's Questions

Comments to the Author

1. Is the manuscript technically sound, and do the data support the conclusions?

Thank you for your constructive comments. Really, we appreciate and accepted all the comments you raised for us. We have made correction to the revised manuscript accordingly.==>please line 48 `and 347.

2. Has the statistical analysis been performed appropriately and rigorously?

Thank you for your constructive comments. We have accepted all the comments you raised. We have made correction based on your suggestion to the revised manuscript. In the revised manuscript statistical analysis have been performed including associated factors and domain of health related quality of life among cancer patients in Ethiopia. ==>Please see line from 281-309.

3. Have the authors made all data underlying the findings in their manuscript fully available?

Thank you for your constructive comments and suggestions. Even if the PLOS Data policy requires authors to make all data underlying the findings described in their manuscript fully available without restriction, the summary statistics, the data points behind means, medians and variance measures were excluded from our study. Because our target was to analysis the pooled prevalence of health related quality of life among cancer patients. In addition to this there is no restrictions on publicly sharing data—e.g. participant privacy or use of data from a third party. Because our study design is systematic review and meta analysis.

On the other hand authors do not need to submit their entire data set and the raw data collected during an investigation. Because both were reported and used in the reported study as well as share data in the main manuscript of this article as the standard in the field that have been processed during analysis.

4. Is the manuscript presented in an intelligible fashion and written in standard English?

PLOS ONE does not copy edit accepted manuscripts, so the language in submitted articles must be clear, correct, and unambiguous. Any typographical or grammatical errors should be corrected at revision, so please note any specific errors here.

Reviewer #1: Yes

Reviewer #2: Yes

Thank you for your constructive comments.

Reviewer #1:  There are several points that make the systematic review fragile, requiring an almost complete restructuring of methodology, results and especially discussion.

Minor questions

1- The submission code on the Prospero platform not found on the platform, as well as the title and keywords.

Thank you so much for your comments. We means that we already registered on the Prospero platform with PROSPERO Registration message with CRD [284157]; , but still no conformation notification is provided. So we accepted your comment 100% and made correction on revised manuscript. ==>Please see line 114.

2- Disagreement between the results found in the PRISMA diagram, in the description of the results and the final number of analyzed articles. 

Thank you for your suggestions and comments. After proof reading, the appropriate correction was made to the revised manuscript based on the given comment to make it consistent throughout the manuscript.

Major questions

- When performing Quality assessment, a SCORE was used, JBIMAStARI does not use a score table, like other qualifiers, so it needs to indicate the use of the Modified Qualifier and as it was estimated that 7 points is a low risk assertive. 

Thank you for your critical comments. The table in our manuscripts helps as a tally sheet to counts the points of critical appraisal of the article. Here the JBI-MAStARI requires for the use as a methodological tool, specifically for assessing risk of bias with 9 items modified quality of life assessment checklist. According to Newcastle-Ottawa quality assessment Scale (NOS) score 7 or more for cross- sectional studies was accepted. Based this a score of 7 or more out of 9 acceptable for this review and meta-analysis. ===>please see line 144, and 177.

2- Discuss how “No” and “Unclear” answers can directly affect the quality and risk of bias of the article.

Thank you for your critical comments. At this manuscript, the methodological qualities of included studies were assessed based on a modified checklist developed to assess the methodological quality aspect of quality of life reporting. A score of yes , no or unclear was given for each item. A score of yes was given for an item if meeting the methodological criteria. A score of no was given for an item is not meeting the methodological criteria, and if an item neither met the criteria nor described the related parameter sufficiently was give unclear. Here , we use the above terms to screen the eligible articles for systematic review and meta-analysis. ===>please see line 172-180.

3-Review the structure of the Meta-analysis, it has an extremely high heterogeneity (i² = 98.8%) which lowers the certainty of the evidence due to data inconsistency.

Thank you for your suggestions. Absolutely! But the reason that the occurrence of high heterogeneity was the number of studies in this review and meta-analysis is small.

4-Demonstrate the concept of Quality of Life and the tools used in the articles in order to seek similarities and differences between them, as well as discuss which domains of quality of life they address.

Thank you very much. We have made the necessary correction. The concept Quality of life is the degree to which an individual is healthy, comfortable, and able to participate in or enjoy life events. It usually used to measure in chronic conditions and frequently impaired to a great extent of the patients. ===>please see line 135 

Concerning measurement tool , Health related quality of life can be measured by patient health questionnaire-9, Caregiver Quality of Life Index-Cancer, European organization for research and treatment of cancer core 30 and quality of life questionnaire specific to breast , European Organization for Research and Treatment of Cancer module and Euro Quality of Life Group’s 5-Domain Questionnaires 5-Levels (EQ-5D) questionnaires were tools that use to differentiated the articles. ==>Please see table 2 and 3. 

Concerning the domain of quality of life, in this manuscript physical functioning, and social functioning significantly associated with quality of life among cancer patients in Ethiopia. ==>Please see line 292 and 297.

5-Characterize the samples of articles taking into account Gender, Age, Socioeconomic level and discuss how this can directly change the certainty of the evidence.

Thank you very much for your constructive comments. We would like to say sorry for the unclear expression. We didn’t address those points in this review and meta-analysis, however we included as a limitation of our study in the revised manuscript. Because all the above mentioned terms are out of our predefined and modified checklist or no primary study concerned for the mentioned variables. Further research is needed to fully assess the impact of variables like Gender, Age, Socioeconomic level on cancer patients. ==>Please see line 357 

6- Discuss the variability of cancer types, if this directly interferes in the context of this review

We would like to say sorry for the unclear expression. Here our concern was not type cancer , but on the pooled prevalence of cancer. So type of cancer is not directly interferes in the context of this review. Here, we show that the lifetime risk of cancers of many different types is strongly associated with the health related quality of life. This is important not only for understanding the disease but also for designing strategies to limit the mortality it causes.

7- Discuss how the age limitation of study participants can be a bias in these studies and thus directly a bias in the statement of the systematic review.

Thank you for your critical comments. We apologize for the ambiguous phrase. Here our concern was on the age category of study participants. All primary articles included in this systematic review and meta analysis have no age category group. As a result, our review could be hampered. 

8-With this review the conclusion, showing evidence of a high quality of life and the possibility of including suggestions for future studies carried out in this area.

With these corrections, resubmit the article for consideration, as it is a topic of extreme relevance.

Great thanks for your valuable suggestions.correction has made accordingly.

Reviewer #2: Abstract section- how did you assess health related quality of life among cancer patients using pooled prevalence? since quality of life is assessed on the basis of score. 

Great thanks for your critical comment. We apologize for the ambiguous phrase. Corrections have made accordingly. After proof reading, the appropriate correction was made to the revised manuscript based on the given comment to make it consistent throughout the manuscript.

We corrected the term with pooled estimates mean score of health related quality of life among cancer patients based on standard tool results thorough the revised manuscript. 

Method section- what was the rationale for including various study designs like case control and cohort studies.

which study tool was used to assess health related quality among cancer patients? this has not been mentioned anywhere in methodology.

Thank you very much for your valuable comment. Here, the rational of using various study designs like case control and cohort studies were if the outcome variable of the previous primary study was similar. 

Concerning assessment tool of health related quality of life was measured by using WHOQOL-HIV BREF. ==>Please see line 243 and table 3

Result section: somewhere it is mentioned 13 studies were included but in tables there are only 12 studies.

Also, there is no homogeneity among cancers in different studies. some studies have included only breast cancer, some have included included only cervical cancers and some have included all cancers. Therefore the results cannot be generalized to overall health related quality among all cancer patients. kindly justify ?

Thank you for your critical view and we would like to say sorry for the error. After proof reading, the appropriate correction was made to the revised manuscript based on the given comment to make it consistent throughout the manuscript.

Thank you very much for your critical suggestion concerning to generalization. Here our goal is’’ to estimate the overall mean score of health-related quality of life’’ among any type of cancer patients because our source of population is made up of cancer patients in Ethiopia. 

6.PLOS authors have the option to publish the peer review history of their article (what does this mean?). If published, this will include your full peer review and any attached files.

It means PLOS now offers accepted authors the opportunity to publish the peer review history of their manuscript alongside the final article. The peer review history package includes the complete editorial decision letter for each revision, with reviews, and your responses to reviewer comments, including attachments.

---

## [Editor Report · Decision Letter 2]

4 Nov 2022

Health-related quality of life and associated factors among cancer patients in Ethiopia: Systematic review and Meta-analysis

PONE-D-21-39423R2

Dear Dr.Tadele

We’re pleased to inform you that your manuscript has been judged scientifically suitable for publication and will be formally accepted for publication once it meets all outstanding technical requirements.

Kind regards,

José Luiz Fernandes Vieira

Academic Editor

PLOS ONE

Additional Editor Comments (optional):

Dear Dr. Tadele

All issues of the reviewers were adeqautely ansewered by authors. Congratulations for the changes in the manuscript

best regards

josé luiz vieira

All the suggestions were included in the manuscript

---

## [Editor Report · Acceptance letter]

9 Nov 2022

PONE-D-21-39423R2 

Health-related quality of life and associated factors among cancer patients in Ethiopia: Systematic review and Meta-analysis 

Dear Dr. Lankrew Ayalew:

I'm pleased to inform you that your manuscript has been deemed suitable for publication in PLOS ONE. Congratulations! Your manuscript is now with our production department. 

Kind regards, 

on behalf of

Dr. José Luiz Fernandes Vieira 

Academic Editor

PLOS ONE